# The Influence of Perceived External Prestige on Emotional Labor of Frontline Employees: The Mediating Roles of Organizational Identification and Impression Management Motive

**DOI:** 10.3390/ijerph191710778

**Published:** 2022-08-30

**Authors:** Pengfei Cheng, Jingxuan Jiang, Zhuangzi Liu

**Affiliations:** School of Economics and Management, Xi’an University of Technology, Xi’an 710054, China

**Keywords:** perceived external prestige, emotional labor, organization identification, impression management motive, perceived organizational support

## Abstract

Drawing on both the organization identification and impression management theories, we propose that perceived external prestige of frontline employees influences their emotional labor through organizational identification and impression management motive. Further, the relative influence of either pathway depends upon perceived organizational support. Using survey data from 377 frontline employees in 104 hotels, the results indicate that perceived external prestige is positively related to deep acting, and negatively related to surface acting. Organizational identification partially mediates the relationship between perceived external prestige and deep acting. However, the relationship between perceived external prestige and surface acting is partially mediated both by organizational identification and impression management motive. In addition, perceived organizational support positively moderates the relationship between perceived external prestige and organizational identification, and negatively moderates the relationship between perceived external prestige and impression management motive, respectively.

## 1. Introduction

As competition becomes fiercer, service firms rely more on providing an excellent service experience to survive. During service encounters, how frontline employees regulate and display emotions is so important in shaping customers’ service experience that almost all service firms take “service with a smile” as a very important mantra [1]. Hochschild [2] defined emotional labor as how employees regulate and display their emotions during service encounters to conform with organizations’ display rules. Although there are two different strategies employees can take to engage in emotional labor, surface acting and deep acting [2,3], the outcomes of deep acting and surface acting are different. Deep acting refers to employees trying to display appropriate emotions by actually experiencing or feeling the emotions; it contributes to positive outcomes such as customer emotional experience [4], employee job satisfaction [5], and perceived service quality [6]. Surface acting refers to employees feigning emotions that are not actually felt; it causes negative outcomes such as burnout [7], negative customer emotion [8], and negatively affecting customers’ subjective perception [9].

Given the pivotal role of emotional labor in shaping service experience [6], both academics and practitioners are increasingly interested in how to effectively manage frontline employees’ emotional labor. However, considering the dynamism and complexity of service interactions, it is impossible for managers to monitor whether an employee engaged in deep or surface acting during the service encounters. Frontline employees’ emotional labor, to a great extent discretionary behaviors [10], goes beyond their supervisor’s direct control and are mainly driven by employees’ intrinsic motivation [11]. Hence, it is important for service firms to explore factors that can serve as intrinsic motivations of frontline employees’ emotional labor and understand the mechanisms by which the influence was exerted.

Smidts, et al. [12] defined perceived external prestige (PEP) as an employee’s perception of how outsiders judge the status and image of his or her organization. According to social identity theory, an individual may generate strong attachment, pride, and self-enhancement motive to his or her organization that holds great reputation [13]. Previous research suggests that PEP plays a pivotal role in employees’ attitudes and behavior, especially their discretionary actions (e.g., organizational citizenship behavior) [13,14]. Yet there is little research regarding the influence of PEP on the emotional labor of frontline employees [15]. In addition, as an important organizational phenomenon, impression management has a critical influence on employees and their firms. For example, impression management motive fosters employees’ organizational citizenship behavior and positivity [16,17]. Emotional display, as one type of employee impression management behavior [18], can be motivated by impression management motivation. However, researchers to date have paid scant attention to the effect of PEP on employees’ behavior from an impression management perspective. Therefore, the purpose of this research is to explore the relationship between PEP, impression management motive, organizational identification, and employees’ emotional labor.

Drawing upon the impression management theory and social identity theory, we propose that PEP influences employees’ emotional labor through dual mediating processes, namely organizational identification and impression management motive. Furthermore, perceived organizational support (POS) affects employees’ beliefs concerning their legitimacy as organizational members [19,20]. The external prestige of the organization is salient to employees only when they feel themselves to be legitimate organizational members [21]. Thus, we also highlight the moderating role of POS, which may alter the relationship between PEP and employees’ emotional labor. This research may extend previous literature on emotional labor and perceived external prestige by opening the “black box” between PEP and emotional labor and considering the boundary conditions of these relationships.

## 2. Theoretical Background and Hypothesis Development

### 2.1. Emotional Labor

Emotional labor is defined as behaviors by frontline employees for displaying appropriate emotions to conform to organizational display rules [2]. Previous studies have described two dimensions of emotional labor strategies: surface acting and deep acting [2,3]. Specifically, surface acting involves a faking process, where employees express the expected emotion without necessarily altering how they feel. In contrast, deep acting is described as a more effortful process, where employees try to display expected emotions by adjusting their inner feelings [3]. Due to the importance of emotional labor in the service experience, in recent years, considerable research has tried to help service firms control and manage frontline employees’ emotional labor by identifying antecedents from an organizational perspective, such as organizational justice [22], organizational dehumanization [23], and leadership [24]. As a discretionary behavior, frontline employees’ emotional labor is prone to be driven by their intrinsic motivations. However, the mechanisms by which these organizational factors internalize and influence frontline employees’ emotional labor have received little explicit attention in organizational scholarship. Hence, addressing the antecedents to emotional labor from an organizational perspective and unveiling the process of these effects may enable service firms to manage employees’ emotional labor more effectively, and consequently create excellent service experience.

### 2.2. Perceived External Prestige

Perceived external prestige (PEP) refers to the employee’s individual beliefs about how the external relevant stakeholders, such as customers, competitors, and suppliers, view his/her organization [12,25]. It is important to note that external prestige and organizational reputation are closely associated, but two distinct constructs. Organizational reputation refers to the overall assessment of current organizational assets, market position, and future behavior [26], which reflects outsiders’ beliefs. In contrast, PEP reflects insiders’ (e.g., employees) beliefs about their organization. For the same organization, the cognitions or evaluations of outsiders and insiders may result from different sources of information [12]. Thus, even though an organization’s external prestige is often related to its reputation, they are two different constructs.

#### 2.2.1. The Mediating Role of Organizational Identification

Prior research on external prestige suggests that the effects of PEP on employees’ attitudes and behavior toward their organization are mediated by organizational identification [12]. As a psychological foundation of the employee–organization relationship, organizational identification refers to the extent that an employee’s perception of oneness with or belonging to her or his organization [21]. PEP reflects how the public view the organization, and is acknowledged as an antecedent of organizational identification [27]. Driven by employees’ self-enhancement motive, high external prestige not only enhances the attractiveness of organizational membership to an employee but ultimately results in proactive behaviors [28]. During service encounters, frontline employees, who act as representatives of the service firm in interactions with customers, are likely to depersonalize the self and use the organization as a vehicle for self-definition [29]. Under this condition, organizations with high external prestige can enhance or maintain employees’ positive social identity [30]. Thus, PEP contributes to enhancing employees’ self-esteem and organization identification.

**Hypothesis** **1.** 
*Perceived external prestige is positively related to organizational identification.*


Given the discretionary nature of frontline employees’ emotional labor [10], it is logical to expect that frontline employees’ deep acting or surface acting during the service encounter is driven by their intrinsic motivation (e.g., organizational identification) [31]. According to social identity theory, employees who strongly identify with their firms are prone to take organizational goals as their own and consequently engage in behaviors that are beneficial to the firms [32]. For instance, employees with high organizational identification are more likely to engage in OCB [33] and customer orientation behaviors [34], which are especially important in service industries. Therefore, we can expect that, during service encounters, frontline employees with high organizational identification would internalize organizational display rules and devote more effort to altering their inner feelings to obey the display rules, namely engaging in deep acting [35]. Conversely, employees with low organizational identification would ignore organizational interests and display rules by merely pretending “fake” emotions, namely conducting surface acting.

**Hypothesis** **2.** 
*Organizational identification is positively related to deep acting.*


**Hypothesis** **3.** 
*Organizational identification is negatively related to surface acting.*


#### 2.2.2. The Mediating Role of the Impression Management Motive

Impression management theory suggests that people care about how they are viewed by others [16,36]. To shape a new or maintain a current personal image, employees manage their image that is projected to the target population (e.g., interviewers, supervisors, or customers) through strategic behavior at work [37,38]. Impression management processes are generally conscious and tactical because employees are prone to shape or maintain specific images [39].

Research on impression management has typically focused on the internal context of organizations, such as how employees affect the personal perceptions of their colleagues, supervisors, and subordinates of them through impression management [16,40,41]. Although the targets of employees’ impression management also include their customers [42], few studies pay attention to frontline employees’ impression management when interacting with customers. Yun et al. [43] defined an individual’s impression management motive as the desire to enhance one’s self-image by consciously exhibiting specific behaviors. When frontline employees, as a service firm’s representatives, interact with customers, the external prestige of the employees’ organization becomes an important and valued component of their self-image. Therefore, frontline employees have strong motive to use impression management strategies to maintain their positive self-image, which is derived from their organization’s prestige. Thus, we propose:

**Hypothesis** **4.** 
*Perceived external prestige is positively related to impression management motive.*


As representatives of service firms, frontline employees deliberately attempt to regulate and display their emotions (emotional labor) during the service encounter, in order to shape customers’ perception of themselves and the firms. Therefore, Ashforth and Humphrey [44] argue that emotional labor can be considered a form of impression management. There are many common skills which can be used for both emotional labor and impression management. Bolino, Long, and Turnley [16] indicate that smiling and eye contact may be both important impression management tactics and salient ways for employees to display emotions during service interactions. Verbal greetings and farewells not only are an important way for frontline employees to express positive emotions [16], but also symbolize employees’ willingness to build, maintain, and enhance customer relationships, which is similar to ingratiation tactics in impression management. Thus, we can expect that frontline employees’ emotional labor would be driven by an impression management motive.

In general, people’s impression management is conscious and tactical [39]. Therefore, previous research has suggested that the image a person projects to others through impression management behaviors is perhaps not authentic, but rather fake [38], which is a similarity between impression management and employee surface acting. Ashforth and Humphrey [44] mentioned that surface acting, which focuses directly on one’s outward behavior, was the form of acting typically discussed as impression management. In contrast, deep acting focuses on one’s inner feelings, beyond the notion of impression management. In other words, driven by impression management motives, employees’ positive displays may be due to maintaining or enhancing the image they project to customers, rather than expressing authentic emotions they feel. We can also get similar statements from Albrecht et al. [45], who argued that surface acting is an important impression management tactic, and Shumski Thomas et al. [46], who found that many employees hide authentic emotions by surface acting to avoid offending their supervisors and maintain a good image. Thus, we propose that employees, driven by impression management motives, are prone to engage in more surface acting and less deep acting during service interactions.

**Hypothesis** **5.** 
*Impression management motive is negatively related to deep acting.*


**Hypothesis** **6.** 
*Impression management motive is positively related to surface acting.*


### 2.3. The Moderating Role of Perceived Organizational Support

People generally are motivated to improve and maintain their status and self-worth [47]. Frontline service employees, as boundary-spanners, interact not only with customers outside the firm, but also with supervisors and colleagues inside the organization. Frontline employees’ self-worth therefore depends on both their status during the service encounters and their status in the organization. External prestige reflects the organization’s status, from which frontline employees may inform “how others outside the organization view me”. Thus, high perceived external prestige (PEP) enables frontline employees to internalize a high status when interacting with customers. Perceived organizational support (POS) refers to the perceptions of employees that their organization values and cares about their contributions and well-being [48,49]. As a sign of employees’ status in the organization, POS reflects “how others in the organization (e.g., supervisors or colleagues) view me”. A high level of POS enables employees to experience feelings of respect [14,50].

Previous research suggests that POS affects employees’ beliefs about the legitimacy of their organizational membership [21,51]. When employees’ contributions are valued by an organization, they will perceive they are respected and affectively attached to the organization [48]. Under this condition, employees perceive the legitimacy of their organizational membership. Conversely, if employees perceive less organizational support, they are merely “nominal” rather than legitimate members of the organization. According to social identity theory, employees feel obligated to care about the external status of the organization (external prestige) only when they believe that their organizational membership is legitimate [21,51]. In other words, the external prestige of the firm makes sense and results in employees’ identification only when they perceive a high level of organizational support. In contrast, under the condition of low POS, employees might be unconcerned about the external prestige of the organization, which will undermine the relationship between external prestige and organizational identification.

**Hypothesis** **7.** 
*The positive relationship between perceived external prestige and organizational identification will be moderated by perceived organizational support. Specifically, the relationship will be stronger (weaker) when employees have a high (low) level of perceived organizational support.*


Previous research has demonstrated that PEP and POS are both related to employees’ self-worth. However, these two constructs may be inconsistent for an employee. To enhance external prestige, companies send positive information to external stakeholders via marketing communication [52,53]. However, an employee’s perception of their organization might be different from outsiders’ perception. For example, a firm with high external prestige may surprisingly support its employees less [54]. In this context, although external prestige can make frontline employees feel a high level of self-worth and status when they interact with customers, employees may not generate an attachment to the organization due to the lack of respect and legitimate organizational membership. Thus, frontline employees are more likely to engage in impression management tactics driven by instrumental motivations. In other words, employees with low POS are prone to capture the benefits of PEP (e.g., higher status and self-worth) by deliberately engaging in impression management during the service encounter. Furthermore, employees with low POS doubt the way the organization treats them, and impression management is an effective means for employees to cope with the perception of ambivalence (high external prestige versus low organizational support). Figure 1 presents the research framework.

**Hypothesis** **8.** 
*The positive relationship between perceived external prestige and impression management motive will be moderated by perceived organizational support. Specifically, the relationship will be stronger (weaker) when employees perceive low (high) organizational support.*


## 3. Method

### 3.1. Sample and Procedure

This research selected frontline employees from 104 Chinese hotels for several reasons. First, in response to fiercer competition in the Chinese hospitality industry, external prestige, as one of the intangible assets, has become an important means for hotels to establish their sustainable competitiveness. Second, various tiers of hotels provide sufficient variation in external prestige for this study. Finally, almost all hotels have emotional requirements for frontline employees who are directly in contact with customers during service encounters, which means emotional labor is very common in frontline hotel employees’ daily work.

Questionnaires were sent to 520 frontline employees working at hotels located in Xi’an and Chengdu in China. We selected 80 hotels from four tiers of hotels, with 20 for each tier: economy hotels, three-star hotels, four-star hotels, and five-star hotels. For each hotel, 6–8 frontline employees who directly come in contact with customers were selected to take part in the survey. Finally, a total of 390 questionnaires were collected, with 182 from hotels located in Chengdu and 208 from hotels located in Xi’an, respectively. After deleting 13 incomplete questionnaires, a sample of 377 was used for data analysis, with a response rate of 72.5%. The sample consisted of 27.4% males and 72.6% females.

### 3.2. Measures

For all measures, except for the demographic variables, seven-point Likert-type scales were used ranging from 1 to 7 with 1 indicating “strongly disagree,” and 7 indicating “strongly agree”.

Core variables. The emotional labor of frontline employees was measured with two 3-item subscales adapted from Brotheridge and Lee [55]. One 3-item subscale captured surface acting and the other captured deep acting. Sample items include “Hide my true feelings about a situation” and “Make an effort to actually feel the emotions that I need to display to others”. Organizational identification was measured with the 6-item scale adapted from Homburg, Wieseke, and Hoyer [34], originally developed by Mael and Ashforth [56]. A sample item is “This hotel’s success is my success”. The 6-item scale that was developed by Yun, Takeuchi and Liu [43] was used to measure impression management motive in this research. A sample item was “It is important to me to give a good impression to others”. Concerning perceived external prestige, we adopted a 6-item scale from Herrbach et al. [57]. A sample item was “My organization is considered one of the best”. Perceived organizational support was measured using 6 items by Eisenberger et al. [58]. The items have been used by many studies [59,60].

Control variables. We controlled participants’ gender (0 = male; 1 = female), age, and tenure. Furthermore, frontline employees in top-tier hotels need to obey more stringent emotional display rules and pay more attention to their emotional regulation at work. Thus, we also controlled hotel levels: economy hotels, three-star hotels, four-star hotels, and five-star hotels. Three dummy variables were used for coding hotel levels (five-star: level 1 = 0, level 2 = 0, level 3 = 0; four-star: level 1 = 1, level 2 = 0, level 3 = 0; three-star: level 1 = 0, level 2 = 1, level 3 = 0; economy: level 1 = 0, level 2 = 0, level 3 = 1).

### 3.3. Confirmatory Factor Analysis, Reliability, and Common Method Bias

Confirmatory factor analyses (CFA) were conducted to assess the convergent and discriminant validity of the latent constructs used in this research. The overall fit statistics for hypothesized 6-factor measurement model are acceptable: Chi-square = 1122.13, df = 390, root mean square of approximation (RMSEA) = 0.07, normed fit index (NFI) = 0.93, non-normed fit index (NNFI) = 0.95, comparative fit index (CFI) = 0.95. The factor loadings of all items on their corresponding construct were greater than 0.6 (see Table 1), which indicates the convergent validity was acceptable. For the discriminant validity, as shown in Table 2, the square roots of every construct’s average variance extracted values (AVE) were greater than the corresponding correlation between all constructs, indicating the discriminant validity is acceptable. Finally, acceptable reliability was confirmed by the fact that the Cronbach’s alpha coefficients of all constructs, ranging from 0.781 to 0.920, were above the cutoff value 0.70 as shown in Table 1.

Next, we examined the possible effects of common method bias. A Harman’s one-factor test was used to rule out possible common method variance (CMV) problems [61]. The results show that six factors were extracted when eigen-values were above 1, and the first factor accounted for only 27.12% of the total variance, providing the evidence that CMV was not present.

## 4. Results

### 4.1. Test of the Mediating Effect

To test the mediation effects of organizational identification and impression management motive on the relationships between perceived external prestige (PEP) and emotional labor, we conducted a series of mediation model analyses using PROCESS 3.3 macro for SPSS developed by Hayes [62]. Based on 5000 bootstrapped resamples, PROCESS can provide 95% confidence intervals (CI) for total effects, direct effects, and indirect effects. If the CI excludes zero, the effect is significant. Therefore, it is suitable for testing mediation effects.

First, we tested the effect of perceived external prestige (PEP) on deep acting (DA) via organizational identification (OI) and impression management motive (IMM) by using a simple mediation model from the PROCESS macro. As shown in Table 3 M-1, PEP was positively related to deep acting (β = 0.404, *p* < 0.001). We supposed PEP influences deep acting via organizational identification and impression management motive. Thus, we examined the effects of PEP on organizational identification, in M-2 and impression management motive in M-3, respectively. The results indicated that PEP was positively related to organizational identification (β = 0.617, *p* < 0.001), supporting H1. The results indicate that a firm with a high level of external prestige is more likely to evoke employees’ identification, which is in line with previous research on organizational identification. As shown in M-3, PEP was positively related to impression management motive (β = 0.334, *p* < 0.001) as well, supporting H4. Then, in M-4, both PEP (β = 0.242, *p* < 0.001) and organizational identification (β = 0.256, *p* < 0.001) were positively related to deep acting, supporting H2. However, as the relationship between impression management motive and deep acting was not significant (β = 0.014, *p* > 0.05), H5 was rejected. As shown in Table 4, results of the bootstrapping sample (95% confidence interval) indicated that organization identification (CI = 0.053, 0.179) rather than impression management motive (CI = −0.012, 0.030) mediates the impact of PEP on deep acting. In addition, PEP exerts positive direct effect on deep acting significantly (β = 0.242, *p* < 0.001) (M-4). Taken together, these results confirm that PEP’s effect on deep acting is only partially mediated by organizational identification, rather than by impression management motive.

Second, we tested the effect of perceived external prestige (PEP) on surface acting (SA) via organizational identification (OI) and impression management motive (IMM). As shown in Table 3 M-5, PEP was negatively related to surface acting (β = −0.183, *p* < 0.001). In M-6, the relationship between organizational identification and surface acting was negative and significant (β = −0.253, *p* < 0.001), supporting H3. The effect of impression management motive on surface acting was positive and significant (β = 0.386, *p* < 0.001), supporting H6. After controlling the mediators (organizational identification and impression management motive), PEP significantly exerted a negative effect on surface acting (β = −0.155, *p* < 0.01). As shown in Table 5, results of the bootstrapping sample (95% confidence interval) indicated that both organization identification (CI = −0.196, −0.042) and impression management motive mediated (CI = 0.051, 0.152) the impact of PEP on surface acting. Taken together, these results confirmed that PEP’s effect on surface acting was partially mediated by both organizational identification and impression management motive. A side note of interest is that PEP’s indirect effects on surface acting via organizational identification and impression management motive are negative and positive, respectively. We will discuss these inconsistent mediations later.

### 4.2. Test of the Moderated Mediation

To test the moderating role of perceived organizational support, we conducted moderated mediation analyses using Hayes’ PROCESS macro. In line with Aiken et al.’s [63] guidelines for moderated regression, the predictor (PEP) and moderator (POS) variables were mean-centered before creating the interaction term. As shown in Table 6, the interaction of PEP and POS was positively related to organizational identification (β = 0.083, *p* < 0.05), and the R^2^ change between M-7 and M-8 was significant (ΔR^2^ = 0.008, *p* < 0.01). According to Aiken, West, and Reno [63], we graphed the simple slopes of PEP on organizational identification at high and low level of POS (+1.0, and −1.0 standard deviations from mean), and visualized the form of the moderating effect. As shown in Figure 2, the positive relationship between PEP and organizational identification was stronger among employees with high POS than with low POS. Thus, H7 was supported. To examine the PEP’s conditional indirect effects on deep acting and surface acting via organizational identification, bootstrapping procedures based on 5000 bootstrapped resamples were used to estimate 95% confidence intervals (CI) for PEP-OI-DA and PEP-OI-SA at high and low POS. As shown in Table 7, under the condition of low POS, PEP’s indirect effects on deep acting (indirect effect = 0.091, CI = 0.033, 0.179) and surface acting (indirect effect = −0.090, CI = −0.184, −0.025) via organizational identification were significant. Under the condition of high POS, PEP’s indirect effects on deep acting (indirect effect = 0.138, CI = 0.071, 0.205) and surface acting (indirect effect = −0.136, CI = −0.210, −0.054) via organizational identification were significant as well. 

Then, we tested the moderating effect of POS on the relationship between PEP and impression management motive. As shown in Table 6, the main effect of POS on impression management motive was not significant (β = −0.038, *p* > 0.05) and the interaction of PEP and POS on impression management motive was negative and significant (β = −0.154, *p* < 0.001). The R^2^ change between M-9 and M-10 (ΔR^2^ = 0.024, *p* < 0.001) was significant, supporting H8. We graphed the simple slopes of PEP on impression management motive at high and low levels of POS. Among frontline employees with low POS, the negative relationship between employees’ PEP and impression management motive was stronger than those with high POS (see Figure 3). Because PEP’s indirect effect on deep acting via impression management motive is not significant, we did not report PEP’s conditional indirect effects on deep acting. Using bootstrapping procedures, PEP’s conditional indirect effects on surface acting via impression management motive were examined. As shown in Table 7, under the condition of low POS, PEP’s indirect effects on surface acting (indirect effect = 0.171, CI = 0.112, 0.237) was significant. Under the condition of high POS, PEP’s indirect effect on surface acting (indirect effect = 0.044, CI = −0.003, 0.105) was not significant.

## 5. Discussion

### 5.1. Theoretical Contributions

Exploring the mechanisms by which perceived external prestige (PEP) influences frontline employees’ emotional labor is the purpose of this research. Drawing upon social identity theory and impression management theory, we propose that PEP affects emotional labor via organizational identification and impression management motive. Furthermore, the boundary condition of these mediation effects is examined by introducing perceived organizational support (POS) as a moderator. We contribute to the literature by explaining how and why PEP influence employees’ emotional labor.

In particular, our research contributes to the literature on external prestige and emotional labor. First, we contribute to the understanding of the underlying mechanisms by which PEP influences frontline employees’ behaviors. Although researchers have argued that PEP influences employees’ attitudes and behaviors primarily through organizational identification [13,14], the role of impression management motive has received little attention. We fill this gap by examining the mediation effects of organizational identification and impression management motive simultaneously. It is important to note that PEP’s indirect effects via organizational identification and impression management motive on surface acting are negative and positive, respectively. This inconsistent mediation implies that the mechanisms by which PEP influence surface acting are far more complex than we knew. Either ignoring the mediator or just considering one mediator (organizational identification) will underestimate PEP’s effect on surface acting. Therefore, our research provided a better and more accurate understanding of the relationship between PEP and employees’ emotional labor.

Second, previous research has paid little attention to the conditions under which PEP is more or less likely to influence employees’ attitudes and behaviors. Our research demonstrates that POS strengthens (weakens) the indirect effects of PEP on emotional labor through organizational identification (impression management motive). This contributes to a better understanding of the boundary conditions under which PEP fosters frontline employees’ engaging in deep acting rather than surface acting.

Third, this study enriches existing literature on the antecedents of employees’ emotional labor. Given the importance of employees’ emotional labor in service encounters, exploring antecedents of emotional labor has attracted many researchers’ attention. Compared with the amount of previous research mainly focused on employee perspective [5,64,65], relatively less attention is given to organizational factors. For example, organizational culture and leadership influence employees’ emotional labor [24,66]. Our findings demonstrate that as a form of intangible asset, external prestige can enhance firms’ performance by shaping frontline employees’ emotional labor.

Finally, our research contributes to the impression management literature by introducing impression management motive as a mediator in our theoretical model. Research on impression management suggests that emotion-expressing skills such as smiling and eye contact are important impression management tactics [16,67], while few studies have empirically examined the relationship between impression management and emotional labor. The results suggest that impression management can serve as a motivation that can drive employees’ emotional labor. In addition, different from previous research which focused on impression management in internal organizational contexts, such as interview and supervisor–subordinate relationships [68,69], our research extends impression management theory into the boundary-spanning context, namely frontline employees’ interaction with customers during service encounters.

### 5.2. Managerial Implications

To survive in fiercer competition, service firms are increasingly paying attention to service experiences. The emotions displayed by frontline employees are a key determinant of customer experience in service interactions. Therefore, how to control or intervene in employees’ emotional labor is an important and challenging task for managers. Our findings might shed light on how a service firm can intervene in employees’ emotional labor using intrinsic motivations. Improving external prestige may be an effective means to promote employees’ deep acting. For example, service firms may improve external prestige by communicating the firm’s core values, social responsibility, and excellent financial performance to external stakeholders. However, merely improving external prestige is insufficient. Service firms should also pay attention to employee’s organizational identification and impression management motive. Specifically, managers need to enhance employees’ organizational identification, to guarantee that external prestige can ultimately drive frontline employees to engage in deep acting rather than surface acting. Service firms should also pay attention to POS. In order to improve POS, both managers and colleagues should take measures to support frontline employees. Specifically, managers can change their leadership style, which can evoke employees’ perception of respect and trust in the organization. Service firms should value frontline employees’ contributions by rewarding and caring for their well-being.

### 5.3. Limitations and Future Directions

There are several limitations to this research. First, the sample of this research was collected from the hospitality industry in China. Although the hospitality industry probably is an appropriate setting for our research, collecting data from one industry limits the generalizability of the findings. Thus, future research should explore the effects of PEP on frontline employees’ emotional labor in different service industries. Second, this research employed a retrospective questionnaire, which is a common cross-sectional design in empirical works. However, a comparison of the survey data carried out over several years would be meaningful. Therefore, future researchers should attempt to integrate experimental and longitudinal approaches, which are conducive to making stronger causal inferences. Third, the present study merely examined the relationship between PEP and frontline employees’ emotional labor, while the other consequences of employees’ positive behaviors might also be meaningful, such as job crafting. Finally, this research merely focuses on the moderating effect of POS. Future research should investigate other moderators’ roles in the relationship between PEP and impression management motive, such as customer orientation.

## 6. Conclusions

To examine the mediating effect of organizational identification and impression management motivation on the relationship between PEP and emotional labor, as well as the moderating effect of POS, we conducted a series of mediation model analyses and moderated mediation analyses using Hayes’ PROCESS macro. Using survey data from 377 frontline employees in 104 Chinese hotels, the results demonstrate that PEP’s effects on deep acting are partially mediated by organizational identification. The influence of PEP on surface acting was partly mediated both by organizational identification and impression management motive. A side note of interest is that the indirect effect of PEP on surface acting through organizational identification is negative, while it is positive via impression management motive. Thus, the total effect of PEP on surface acting was underestimated if the mediating role of organizational identification and impression management motive were not considered. Furthermore, the results also suggest that POS positively moderates the influence of PEP on organizational identification, while it negatively moderates the effect of PEP on impression management motive. In other words, for employees with high POS, PEP is more likely to influence their emotional labor via organizational identification, while for employees with low POS, PEP influences frontline employees’ emotional labor primarily through impression management motive.

## Figures and Tables

**Figure 1 ijerph-19-10778-f001:**
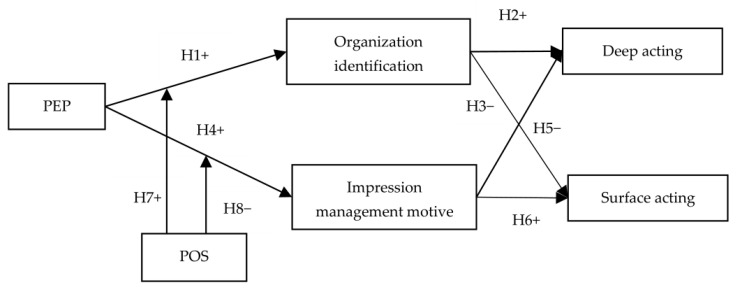
Conceptual model.

**Figure 2 ijerph-19-10778-f002:**
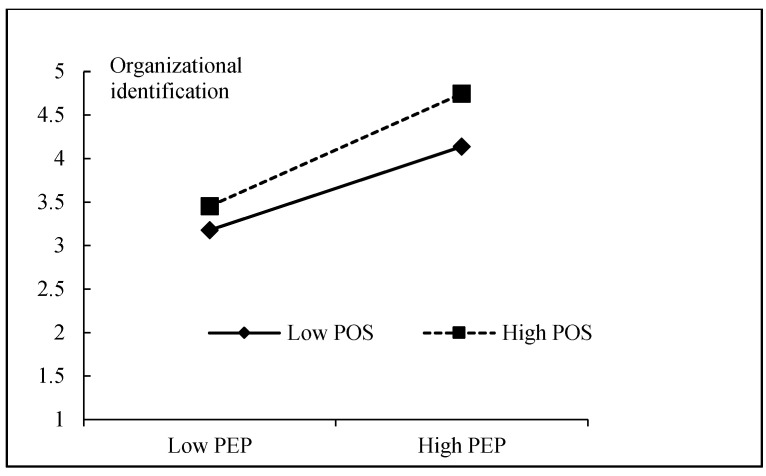
The moderating effect of POS on the relationship between PEP and organizational identification.

**Figure 3 ijerph-19-10778-f003:**
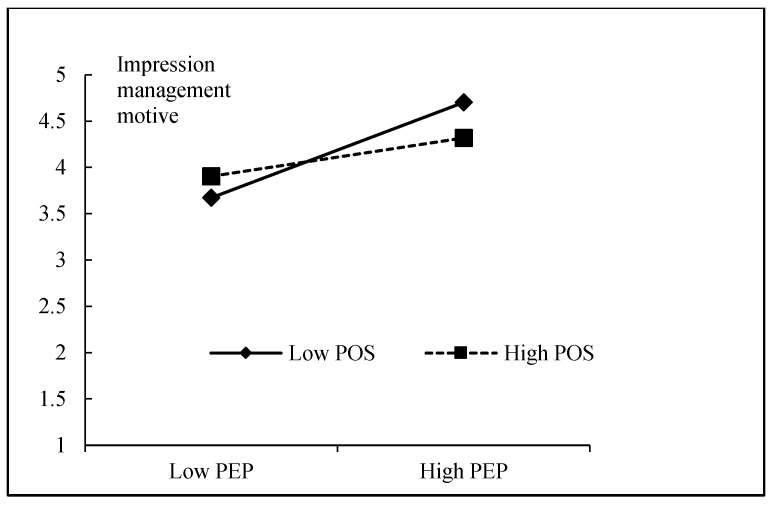
The moderating effect of POS on the relationship between PEP and impression management motive.

**Table 1 ijerph-19-10778-t001:** CFA and Reliability.

Variables	Items	Factor Loadings	T	Variables	Items	Factor Loadings	T
Deep acting α = 0.781	DA-1	0.68	13.50	Surface acting α = 0.866	SA-1	0.81	18.06
	DA-2	0.78	15.98		SA-2	0.84	18.92
	DA-3	0.76	15.65		SA-3	0.83	18.39
OI α = 0.903	OI-1	0.78	17.39	IMM α = 0.908	IM-1	0.82	19.06
	OI-2	0.79	17.71		IM-2	0.81	18.46
	OI-3	0.79	17.85		IM-3	0.80	18.31
	OI-4	0.83	19.22		IM-4	0.81	18.55
	OI-5	0.81	18.55		IM-5	0.79	17.74
	OI-6	0.69	14.66		IM-6	0.70	15.12
PEP α = 0.920	PEP-1	0.78	17.60	POS α = 0.876	POS-1	0.73	15.83
	PEP-2	0.81	18.52		POS-2	0.82	18.54
	PEP-3	0.85	20.22		POS-3	0.79	17.77
	PEP-4	0.79	17.95		POS-4	0.83	18.90
	PEP-5	0.83	19.49		POS-5	0.62	12.63
	PEP-6	0.81	18.74		POS-6	0.62	12.82

Note: OI = organization identification; PEP = perceived external prestige; IMM = impression management motive; POS = perceived organizational support.

**Table 2 ijerph-19-10778-t002:** Correlation matrix and AVE.

	*Means*	*SD*	1	2	3	4	5	6
1. Deep acting	4.38	1.053	0.741					
2. Surface acting	4.34	1.301	−0.223 **	0.843				
3. PEP	4.26	1.328	0.390 **	−0.113 *	0.812			
4. OI	4.16	1.171	0.413 **	−0.262 **	0.515 **	0.783		
5. IMM	4.06	1.166	0.111 *	0.296 **	0.311 **	0.067	0.789	
6. POS	4.35	1.044	0.415 **	−0.130 *	0.208 **	0.277 **	0.073	0.740

Notes: OI = organization identification; PEP = perceived external prestige; IMM = impression management motive; POS = perceived organizational support. The square roots of AVE are presented in diagonal elements (bold values). * *p* < 0.05; ** *p* < 0.01.

**Table 3 ijerph-19-10778-t003:** Test of mediating effects.

Independent Variable		M-1	M-2	M-3	M-4	M-5	M-6
		DA	OI	IMM	DA	SA	SA
Control variable	Gender	0.322 ***	0.044	0.012	0.311 ***	0.409 ***	0.416 ***
	Age	0.090	0.113	0.153	0.117	0.255 **	0.167 *
	Tenure	−0.122 **	−0.016	0.085	−0.119 **	0.004	−0.033
	Level 1	−0.039	0.003	0.229	−0.043	−0.075	−0.162
	Level 2	−0.066	−0.049	0.016	−0.054	−0.041	−0.059
	Level 3	−0.323 **	−0.145	−0.077	−0.285 *	−0.165	−0.172
PEP		0.404 ***	0.617 ***	0.334 ***	0.242 ***	−0.183 ***	−0.155 **
OI		--	--	--	0.256 ***	--	−0.253 ***
IMM		--	--	--	0.014	--	0.386 ***
R^2^		0.194 ***	0.281 ***	0.116 ***	0.252 ***	0.056 ***	0.211 ***

Notes: OI = organization identification; PEP = perceived external prestige; IMM = impression management motive; DA = deep acting; SA = surface acting; * *p* < 0.05; ** *p* < 0.01; *** *p* < 0.001.

**Table 4 ijerph-19-10778-t004:** Total, direct, and indirect effects of PEP on deep acting.

		*Effect*	*Boot SE*	*Boot LLCI*	*Boot ULLCI*
Total effects		0.305	0.038	0.230	0.379
Direct effects		0.182	0.045	0.094	0.271
Indirect effects	PEP→OI→DA	0.119	0.035	0.053	0.179
	PEP→IMM→DA	0.004	0.013	−0.012	0.030

Notes: PEP = perceived external prestige; OI = organization identification; IMM = impression management motive; DA = deep acting; SA = surface acting.

**Table 5 ijerph-19-10778-t005:** Total, direct, and indirect effects of PEP on surface acting.

		*Effect*	*Boot SE*	*Boot LLCI*	*Boot ULLCI*
Total effects		−0.138	0.051	−0.238	−0.038
Direct effects		−0.117	0.057	−0.230	−0.004
Indirect effects	PEP→OI→SA	−0.118	0.040	−0.196	−0.042
	PEP→IMM→SA	0.097	0.026	0.051	0.152

Notes: PEP = perceived external prestige; OI = organization identification; IMM = impression management motive; DA = deep acting; SA = surface acting.

**Table 6 ijerph-19-10778-t006:** The moderating effect of perceived organizational support.

Independent Variable		M-7	M-8	M-9	M-10
		OI	OI	IMM	IMM
Control variable	Gender	0.008	−0.005	0.012	0.036
	Age	0.126	0.114	−0.153	−0.132
	Tenure	0.002	0.002	0.085	0.084
	Level 1	0.009	0.000	0.229	0.211
	Level 2	−0.053	−0.040	0.016	−0.008
	Level 3	−0.074	−0.081	−0.078	−0.064
PEP		0.577 ***	0.563 ***	0.334 ***	0.361 ***
POS		0.201 ***	0.221 ***	0.005	−0.038
PEP × POS		--	0.083 *	--	−0.154 ***
R^2^/ΔR^2^		0.308 ***/--	0.316 ***/0.008 **	0.116 ***/--	0.140 ***/0.024 ***

Notes: PEP = perceived external prestige; OI = organization identification; IMM = impression management motive; POS = perceived organizational support; * *p* < 0.05; ** *p* < 0.01; *** *p* < 0.001.

**Table 7 ijerph-19-10778-t007:** Results for conditional indirect effects.

Mediators		Indirect Effect	Boot SE	Boot LLCI	Boot ULCI
PEP→OI→DA	POS-1SD	0.091	0.038	0.033	0.179
	POS+1SD	0.138	0.034	0.071	0.205
PEP→OI→SA	POS-1SD	−0.090	0.041	−0.184	−0.025
	POS+1SD	−0.136	0.040	−0.210	−0.054
PEP→IMM→SA	POS-1SD	0.171	0.032	0.112	0.237
	POS+1SD	0.044	0.028	−0.003	0.105

Notes: PEP = perceived external prestige; OI = organization identification; IMM = impression management motive; DA = deep acting; SA = surface acting; POS = perceived organizational support.

## Data Availability

Not applicable.

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
