# Peer review of "The Influence of Perceived External Prestige on Emotional Labor of Frontline Employees: The Mediating Roles of Organizational Identification and Impression Management Motive"

_ijerph, 2022, doi:10.3390/ijerph191710778_

Round 1

Reviewer 1 Report

Article : How Does Perceived External Prestige Influence Emotional Labor? The Mediating Roles of Organization Identification and Impression Management Motive

The strengths of the research are:

1.The originality of the topic

2. Accurate way of presenting the results

3. Clear presentation of research hypotheses and reference to them in the analysis of the results.

Weaknesses:

1. Selection of only one industry

2. No repeatability of the tests (a comparison of the results carried out over several years would provide additional information

3. Limited spatial selection of research

Author Response

Dear Reviewer 1:

Thank you for giving us the opportunity to submit a revised draft of our manuscript titled “The Influence of Perceived External Prestige on Emotional Labor of Frontline Employees: The Mediating Roles of Organizational Identification and Impression Management Motive” (Manuscript ID: ijerph-1877999). We appreciate your time and efforts dedicated to providing your insightful comments on our manuscript. We have been able to incorporate changes to reflect most of the suggestions. We have used the “Track Changes” function to mark the revisions within the manuscript.

Here is a point-by-point response to the comments and concerns.

Point 1: Selection of only one industry

Response 1: Thank you for pointing this out. Given the large difference between different industries, frontline employees from different industries show different intent, frequency of emotional labor. Selecting one industry(restaurant, insurance, banking, hotel et al.) to collect data is effective to avoid industry’s effects on emotional labor. It is common in literature on emotional labor. Therefore, we firstly explained the reasons for choosing the hospitality industry as the research context, and secondly added this limitation in the Limitations and Future Directions section.

  • Page 6, line 315: “This research selected frontline employees from 104 Chinese hotels for several reasons. First, in response to the fiercer competition in Chinese hospitality industry, external prestige, as one of the intangible assets, has become an important means for hotels to establish their sustainable competitiveness. Second, various tiers of hotels provide sufficient variation in external prestige for this study. Finally, almost all hotels have emotional requirements for frontline employees who contact customers directly during service encounters, which means emotional labor is very common in hotel frontline employees’ daily work.”
  • Page 13, line 745: “First, the sample of this research was collected from the hospitality industry in China. Although the hospitality industry probably is an appropriate setting for our research, collecting data from one industry limits the generalizability of the findings. Thus, future research should explore the effects of PEP on frontline employees’ emotional labor in different service industries.”

Point 2: No repeatability of the tests (a comparison of the results carried out over several years would provide additional information)

Response 2: Thank you for pointing this out. We have revised the Limitations and Future Directions section to acknowledge this weakness as follows.

  • Page 13, line 748: Second, this research employed retrospective questionnaire, which is a common cross-sectional design in empirical works. However, a comparison of the survey data carried out over several years would be meaningful. Therefore, future researchers should attempt to incorporate experimental and longitudinal approaches, which are conducive to making stronger causal inferences.

Point 3: Limited spatial selection of research

Response 3: We have revised the Limitations and Future Research section to address this issue by providing more future research directions that could be considered.

  • Page 13, line 753: “Third, the present study merely examined the relationship between PEP and frontline employees’ emotional labor, while the other consequences of employees’ positive behaviors might also be meaningful such as job crafting. Finally, this research merely focuses on the moderating effect of POS. Future research should investigate other moderators’ roles in the relationship between PEP and impression management motive such as customer orientation.”

We look forward to hearing from you in due time regarding our submission and to responding to any further questions and comments you may have.

Sincerely,

Authors

Reviewer 2 Report

I begin by congratulating the authors and researchers of the manuscript intitled “How Does Perceived External Prestige Influence Emotional Labor? The Mediating Roles of Organization Identification and Impression Management Motive”. The topic is actual and very interesting.

Personally, I disagree with the actual title. As suggestion, something like “The influence of Perceived External Prestige on Emotional Labor by the Mediating Roles of Organization Identification and Impression Management Motive in Chinese hospitality” could be more informative and clearer.

The article is very well written, and the results are worthy of publication. Check only the format of titles and subtitles as there are no spaces between the numbering and the text.

There is no information about software used for analysis: AMOS, smartPLS or R ?

Dummy variables for hotel level seems incomplete (line 280, 281, 282): where are 5 stars coding?

About your mediation and moderation approaches (following Baron and Kenny), I would recommend you use the macro of Hayes (known as Process, a script for SPSS).

Good luck!

Author Response

Dear Reviewer 2:

Thank you for giving us the opportunity to submit a revised draft of our manuscript titled “The Influence of Perceived External Prestige on Emotional Labor of Frontline Employees: The Mediating Roles of Organizational Identification and Impression Management Motive” (Manuscript ID: ijerph-1877999). We appreciate your time and efforts dedicated to providing your insightful comments on our manuscript. We have been able to incorporate changes to reflect most of the suggestions. We have used the “Track Changes” function to mark the revisions within the manuscript.

Here is a point-by-point response to the comments and concerns.

Point 1: Personally, I disagree with the actual title. As suggestion, something like “The influence of Perceived External Prestige on Emotional Labor by the Mediating Roles of Organization Identification and Impression Management Motive in Chinese hospitality” could be more informative and clearer.

Response 1: Thank you for your suggestion. The actual title has been revised as follows.

  • Page 1, line 2:“The Influence of Perceived External Prestige on Emotional Labor of Frontline Employees: The Mediating Roles of Organizational Identification and Impression Management Motive”

Point 2: The article is very well written, and the results are worthy of publication. Check only the format of titles and subtitles as there are no spaces between the numbering and the text.

Response 2: Thanks for your kind reminders. The spaces between the numbering and the text have been added in titles and subtitles.

Point 3: There is no information about software used for analysis: AMOS, smart PLS or R?

Response 3: We have added content in the results section to address this issue.

  • Page 8, line 384: “To test the mediation effects of organizational identification and impression management motive on the relationships between perceived external prestige (PEP) and emotional labor, we conducted a series of mediation model analyses by using PROCESS 3.3 macro for SPSS developed by Hayes [62].”

Point 4: Dummy variables for hotel level seems incomplete (line 280, 281, 282): where are 5 stars coding?

Response 4: According to dummy variable regression method, if there are N categories of cases in the sample, N-1 dummy variables are needed to represent each category. We have investigated 4 categories hotels: 5-star, 4-star, 3-star and economy. Hence, we set 3 dummy variables(level 1, level 2 and level3) to represent each category. The detailed coding is as following:

5-star: level 1=0, level 2=0, level 3=0;

4-star: level1=1, level 2=0, level 3=0;

3-star: level1=0, level 2=1, level 3=0;

economy: level 1=0, level 2=0, level 3=1

We have revised the Measures section to detaily address this issue.

-Page 6, line 355: “Three dummy variables were used for coding for hotel levels (five-star: level 1=0, level 2=0, level 3=0; four-star: level1=1, level 2=0, level 3=0; three-star: level1=0, level 2=1, level 3=0; economy: level1=0, level 2=0, level 3=1).”

Point 5: About your mediation and moderation approaches (following Baron and Kenny), I would recommend you use the macro of Hayes (known as Process, a script for SPSS).

Response 5: Thank you for your suggestion. We have revised our mediation and moderation approaches by using the macro of Hayes in the Results section.

  • Page 8, line 384: “To test the mediation effects of organizational identification and impression management motive on the relationships between perceived external prestige (PEP) and emotional labor, we conducted a series of mediation model analyses by using PROCESS 3.3 macro for SPSS developed by Hayes [62]. Based on 5000 bootstrapped resamples, the PROCESS can provide 95% confidence intervals (CI) for total effects, direct effects, and indirect effects. If the CI excludes zero, the effect is significant. Therefore, it is suitable for testing mediation effects.”
  • Page 8, line 391: “First, we tested the effect of perceived external prestige (PEP) on deep acting (DA) via organizational identification (OI) and impression management motive (IMM) by using a simple mediation model from PROCESS macro……”

We look forward to hearing from you in due time regarding our submission and to responding to any further questions and comments you may have.

Sincerely,

Authors
